# Deep Learning Generalization
# and the Convex Hull of Training Sets

**Roozbeh Yousefzadeh**
Yale University and VA Connecticut Healthcare System
roozbeh.yousefzadeh@yale.edu

## Abstract

In this work, we study the generalization of deep learning functions in relation to the convex hull of their training sets. A trained image classifier basically partitions its domain via decision boundaries, and assigns a class to each of those partitions. The location of decision boundaries inside the convex hull of training set can be investigated in relation to the training samples. However, our analysis shows that in standard image classification datasets, most testing images are considerably outside that convex hull. Therefore, the performance of a trained model partially depends on how its decision boundaries are extended outside the convex hull of its training data. From this perspective, over-parameterization of deep learning models may be considered a necessity for shaping the extension of decision boundaries. At the same time, over-parameterization should be accompanied by a specific training regime, in order to yield a model that not only fits the training set, but also its decision boundaries extend desirably outside the convex hull. To illustrate this, we investigate the decision boundaries of a neural network, with various degrees of over-parameterization, inside and outside the convex hull of its training set. Moreover, we use a polynomial decision boundary to study the necessity of over-parameterization and the influence of training regime in shaping its extensions outside the convex hull of training set.

## 1  Introduction

A deep learning image classifier is a mathematical function that maps images to classes, i.e., a deep learning function [Strang, 2019]. These models/functions have shown to be exceptionally useful in real-world applications. However, generalization of these functions is considered a mystery by deep learning researchers [Arora et al., 2019]. These models have orders of magnitude more parameters than their training samples [Belkin et al., 2019, Neyshabur et al., 2019], and they can achieve perfect accuracy on their training sets, even when the training images are randomly labeled, or the contents of images are replaced with random noise [Zhang et al., 2017]. The training loss function of these models has infinite number of minimizers, where only a small subset of those minimizers generalize well [Neyshabur et al., 2017a]. If one succeeds in picking a good minimizer of training loss, the model can classify the testing images correctly, nevertheless, for any correctly classified image, there are infinite number of images that look the same, but models will classify them incorrectly (phenomenon known as adversarial vulnerability) [Papernot et al., 2016, Shafahi et al., 2019, Tsipras et al., 2019]. Here, we study some geometric properties of standard training and testing sets to provide new insights about what a model can learn from its training data, and how it can generalize.

Specifically, we study the convex hulls of image classification datasets (both in the pixel space and in the wavelet space), and show that most of testing images fall outside the convex hull of training sets, with various distances from the hull. We investigate the perturbations required to bring the testing

Deep Learning through Information Geometry Workshop
34th Conference on Neural Information Processing Systems (NeurIPS 2020), Vancouver, Canada.

images to the surface of convex hull and observe that the directions to convex hulls contain valuable information about images, corresponding to distinctive features in them. We also see that reaching the convex hull significantly affects the contents of images. Therefore, the performance of a trained model partially depends on how well it can extrapolate. We investigate this extrapolation in relation to the over-parameterization of neural networks and the influence of training regimes in shaping the extensions of decision boundaries.

## 2 Geometry of testing data w.r.t the convex hull of training sets

First, we show that for standard datasets: MNIST [LeCun et al., 1998] and CIFAR-10 [Krizhevsky, 2009], most of their testing data are outside the convex hull of their training sets. We denote the convex hull of a training set by $\mathcal{H}^{tr}$.

To verify whether an image/data point is inside its corresponding $\mathcal{H}^{tr}$ or not, we can simply try to fit a hyper-plane separating the point and the training set. If we find such hyper-plane, the point is outside the convex hull, and vice versa. This is basically a linear regression problem and there are many efficient and fast methods to perform it, e.g., [Goldstein et al., 2015]. For the MNIST dataset, we see that about 95.1% of testing images are outside the $\mathcal{H}^{tr}$, in the pixel space. For CIFAR-10, that percentage is more than 99.9%. When we transform the images with wavelets (an operation analogous to convolutional neural nets), these percentages almost remain the same.

We can now investigate the testing data outside the $\mathcal{H}^{tr}$, to see how far they are located from it. For every testing image, $x_i^{te}$, that is outside the $\mathcal{H}^{tr}$, we would like to find the closest point to it on the $\mathcal{H}^{tr}$, and we denote that point by $x_i^{\mathcal{H}}$. We are interested to know the direction of the shortest vector that can bring the testing image to $\mathcal{H}^{tr}$. We are also interested in how far the testing images are from the $\mathcal{H}^{tr}$.

### 2.1 How far are testing images from the $\mathcal{H}^{tr}$?

The point on the $\mathcal{H}^{tr}$, closest to a point outside it, is the solution to a well defined optimization problem, consisting of a linear least squares objective function and linear constraints. The objective function seeks to minimize the distance between $x_i^{te}$ and $x_i^{\mathcal{H}}$, while the linear constraints ensure that $x_i^{\mathcal{H}} \in \mathcal{H}^{tr}$. We note that there are approximation algorithms to solve this problem, e.g., [Blum et al., 2019]. Here, we solve the problem, numerically, using the gradient projection algorithm described by [Nocedal and Wright, 2006, Chapter 16]. Figure 1 shows the histogram of distance to $\mathcal{H}^{tr}$ for the testing images of the above datasets.

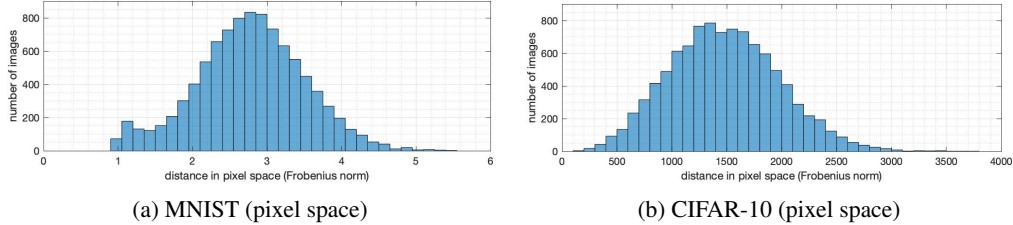

(a) MNIST (pixel space)  (b) CIFAR-10 (pixel space)

Figure 1: Variations of distance to $\mathcal{H}^{tr}$ for testing images that fall outside $\mathcal{H}^{tr}$.

To get a better sense of how far these distances are, consider the $\mathcal{H}^{tr}$ of CIFAR-10 dataset. Its diameter, the largest distance between any pair of vertices in $\mathcal{H}^{tr}$, is 13,621 (measured by Frobenius norm in pixel space). On the other hand, the distance of farthest testing image from the $\mathcal{H}^{tr}$ is about 3,500 (about 27% of the diameter of $\mathcal{H}^{tr}$). Moreover, the average distance between pairs of images in the training set of CIFAR-10 is 4,838, while the closest pair of images are only 701 apart.

Hence, the distance of testing data to $\mathcal{H}^{tr}$ is not negligible and we cannot dismiss it as a small noise. However, it is not very large either. Overall, we can say that in order to classify most of the testing images in the above datasets, a model has to extrapolate, to some moderate degree, outside its $\mathcal{H}^{tr}$. In Appendix A, we discuss the convex hull of random points in high-dimensional space.

## 2.2 Directions to $\mathcal{H}^{tr}$ and the information they contain

For each image that is outside the $\mathcal{H}^{tr}$, there is some minimum perturbation that would bring that image to the $\mathcal{H}^{tr}$. Figures 2 and 3 show the perturbation for some images in the testing set of CIFAR-10 and MNIST that can bring them to their corresponding $\mathcal{H}^{tr}$. We note that due to our approximation method, there is no guarantee that images in the middle column are exactly the minimum required perturbation, but we expect it to be sufficiently close to that minimum.

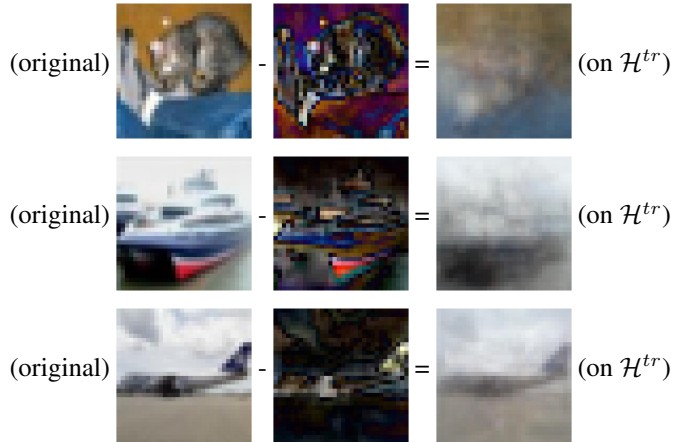

Figure 2: Perturbation (close to minimum) that can bring a testing image to $\mathcal{H}^{tr}$ of all classes. (left image) original testing image from CIFAR-10, (middle image) what should be removed from the original image, (right image) the resulting image on the $\mathcal{H}^{tr}$. These directions contain valuable information about the objects depicted in the images.

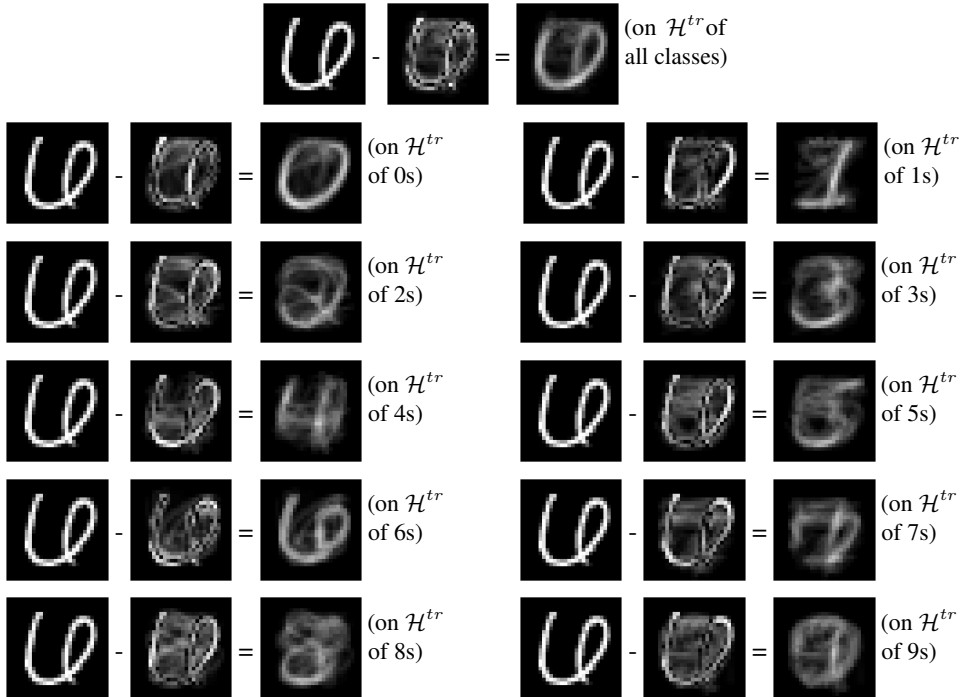

Figure 3: Perturbation that can bring a testing image of MNIST on the $\mathcal{H}^{tr}$.

The perturbations required to bring testing images to the $\mathcal{H}^{tr}$ specifically relate to the objects of interest depicted in images and they appear to impact the images significantly. Therefore, the extrapolation required to classify those images can be considered significant, too.

## 2.3 Images that form the surface closest to the outside image

The image on the surface of $\mathcal{H}^{tr}$, closest to an image, $x_i^{te}$, outside the $\mathcal{H}^{tr}$, is a convex combination of some training images. We call those training images as the support for $x_i^{te}$. In our experiments, the number of support images for each testing image is only a few. For example, the image on the second row of Figure 2 has only 26 support images in the training set. Figure 4 shows 10 of those 26 images with their corresponding coefficients in the convex combination.

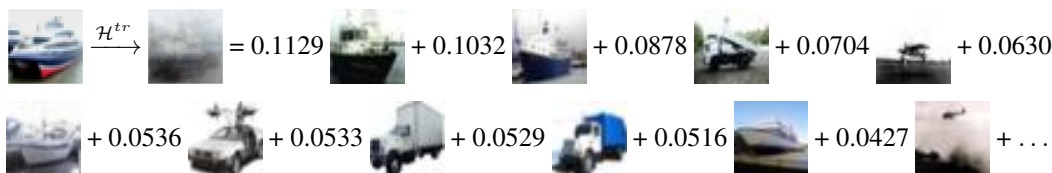

Figure 4: Projection of a testing image to the $\mathcal{H}^{tr}$, and the convex combination of images in training set that create the projected image. There are 26 training images used in the convex combination, but we have shown 10 of those with the largest coefficients.

## 2.4 Related work about geometry of data and deep learning

To the best of our knowledge, convex hulls of training sets are not commonly considered in deep learning studies, especially the ones focused on their generalization. Recently, Yousefzadeh and Huang [2020] reported that in the wavelet space, distance of testing images to the convex hull for each training class can predict the label for more than 98.5% of MNIST testing data. Previously, Haffner [2002] considered the convex hull of MNIST data for Support Vector Machines. Similarly, Vincent and Bengio [2002] considered the convex hulls for K-Nearest Neighbor (KNN) algorithms. However, those methods do not generalize to deep learning functions.

Some researchers have studied other geometrical aspects of deep learning models, e.g., [Cohen et al., 2020, Fawzi et al., 2018, Cooper, 2018, Kanbak et al., 2018, Neyshabur et al., 2017b]. To our knowledge, those studies do not investigate the generalization of deep neural networks in relation to the convex hull of training sets. Most recently, Xu et al. [2020] studied the extrapolation behavior of ReLU perceptrons and concluded that such models cannot extrapolate most non-linear tasks. However, they do not connect their analysis to the fact that a considerable portion of testing samples of standard image datasets fall outside the convex hull of their training sets.

# 3 Learning outside the convex hull: A polynomial decision boundary

In the previous section, we showed that most of the testing data of MNIST and almost all of the testing data of CIFAR-10 are outside the convex hull of their corresponding training sets, while the distance to the $\mathcal{H}^{tr}$ has noticeable variations, resembling a normal distribution. Hence, a trained deep learning model somehow manages to define its decision boundaries accurately enough outside the boundaries of what it has observed during training. But how does a model achieve that, or more precisely, how do we manage to train a model such that its decision boundaries have the desirable form outside the $\mathcal{H}^{tr}$?

Since we are interested in the generalization of image classifiers, and the pixel space is a bounded space, we consider the domain to be bounded, while the $\mathcal{H}^{tr}$ occupies some portion of it. Testing data can be inside and outside the $\mathcal{H}^{tr}$, but always inside the bounded domain.

Let's now use a polynomial decision boundary as an example to gain some intuitive insights.[1] Figure 7a shows two point sets colored in blue and red, each set belonging to a class. These sets are

---

[1]This choice seem appropriate since many recent studies on generalization of deep learning consider the regression models that interpolate, e.g. [Belkin et al., 2018b,a, 2019, Liang et al., 2020, Verma et al., 2019, Kileel et al., 2019, Savarese et al., 2019], but those studies do not consider the convex hull of training sets.

non-linearly separable, because they have no overlap. If we use the polynomial

$$y = 10^{-5}(x + 20)(x + 17)(x + 10)(x + 5)(x)(x - 2)(x - 9), \qquad (1)$$

as our decision boundary, we achieve perfect accuracy in separating these two sets, as shown in Figure 7b.

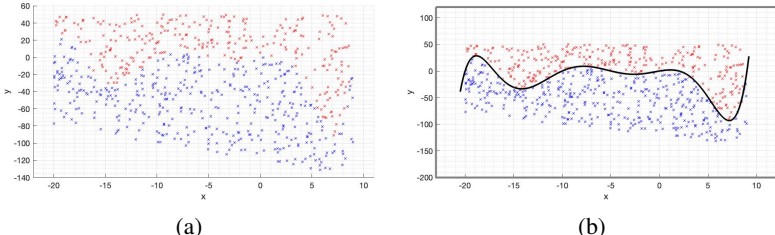

(a)                                                         (b)

Figure 5: **(a)** Training data with 2 classes, colored with blue and red. **(b)** Non-linear separation of 2 classes with a polynomial of degree 7.

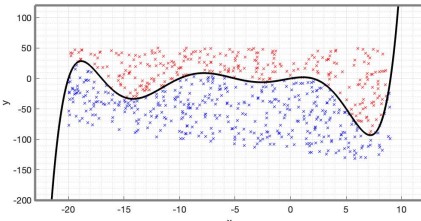

Figure 6: Shape of the polynomial decision boundary in our bounded domain, inside and outside the convex hull of its training data.

Now that we have obtained this polynomial, i.e., decision boundary, we would be interested to know how it generalizes to unseen data. Let's assume that our bounded domain is defined by the limits shown in Figure 6 which also shows how our decision boundary generalizes outside the $\mathcal{H}^{tr}$. If our polynomial can correctly separate and label our testing data, we would say that our polynomial is generalizing well, and vice versa. But, what is reasonable to expect from the testing data? In what regions of the domain should we be hopeful that our polynomial can generalize? What if the domain is much larger than the $\mathcal{H}^{tr}$? Is the extension of our polynomial on both sides reasonable enough?

Clearly, the answer to the above questions can be different inside and outside the $\mathcal{H}^{tr}$. Inside the $\mathcal{H}^{tr}$, if the unseen data has a similar label distribution as the training set, we can be hopeful that our decision boundary will generalize well. However, outside the $\mathcal{H}^{tr}$ is uncharted territory and hence, there will be less hope/confidence about the generalization of our decision boundary, especially when we go far outside the $\mathcal{H}^{tr}$.

Now, let's assume that from some prior knowledge, we know that the decision boundary in Figure 7 is the unique decision boundary that perfectly classifies the testing data. In such case, the decision boundary defined by equation (1) and shown in Figure 6 will generalize poorly outside the $\mathcal{H}^{tr}$, despite the fact that it perfectly fits the training data.

How can we incorporate that prior knowledge into the decision boundary defined by equation (1) and reshape it to the decision boundary in Figure 7, so that it can generalize well both inside and outside the $\mathcal{H}^{tr}$? How can we change the shape of our polynomial outside the $\mathcal{H}^{tr}$, while maintaining its current shape inside the $\mathcal{H}^{tr}$? Clearly, we should add to the degree of our polynomial, or in other words, we should **over-parameterize** it. The necessity of over-parameterization for achieving that goal for our polynomial decision boundary can be rigorously shown using the orthogonal system of Legendre polynomials [Ascher and Greif, 2011].

From this perspective, over-parameterization is necessary, but it is not sufficient for good generalization, because for an over-parameterized polynomial (i.e., decision boundary), there will be infinite number of solutions that can fit the training data, but each of them would have a different shape outside the $\mathcal{H}^{tr}$. In fact, an over-parameterized polynomial can have the same shape as the polynomial

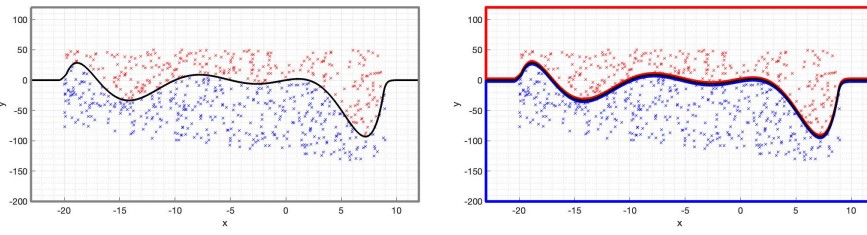

(a) Possible extension of polynomial decision boundary in the over-parameterized regime.

(b) Resulting division of domain between the two classes, defined by red and blue bounds.

Figure 7: Consider the decision boundary depicted in **(a)** and assume that the distribution of testing data is such that the red and blue bounded regions in **(b)** are densely filled with red and blue data points, respectively. It follows that the decision boundary in Figure 6 generalizes poorly for testing points outside the $\mathcal{H}^{tr}$, despite the fact that it perfectly fits the training data.

in Figure 6. But, how can we pick the decision boundary that fits the data and also generalizes well outside the $\mathcal{H}^{tr}$?

In the over-parameterized regime, the key to finding the desirable decision boundary is the optimization process, i.e., **the training regime**. In other words, different training regimes would lead us to decision boundaries that all perfectly fit the training set, but each has a different shape outside the $\mathcal{H}^{tr}$. This highlights that the over-parameterization and the training regime work in tandem to shape the extensions of our decision boundary.

# 4    Output of deep learning functions outside their $\mathcal{H}^{tr}$

In this section, we investigate a 2-layer neural network with ReLU activation functions. We train the model with various number of neurons on the data from previous section as depicted in Figure 7b. We then investigate the output of the trained models inside and outside of the $\mathcal{H}^{tr}$, as shown in Figures 8-10. In these figures, the black trapezoid depicts the $\mathcal{H}^{tr}$. The colors red and blue correspond to our 2 classes.

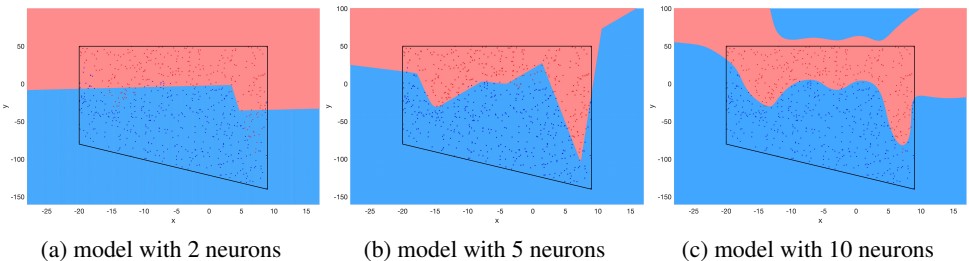

(a) model with 2 neurons      (b) model with 5 neurons      (c) model with 10 neurons

Figure 8: Variations of model output inside and outside the $\mathcal{H}^{tr}$, for under-parameterized models. Feed-forward ReLU networks with 2,5, and 10 neurons do not have enough capacity to perfectly fit the training set. As we increase the number of neurons from 2 to 10, the model fits the training data better, while it starts to have more variations outside the $\mathcal{H}^{tr}$. In this under-parameterized regime, we cannot minimize the training loss to zero, but each time that we train a model, we can achieve the same non-zero training loss for it, leading to the same model.

Because of the non-convexity of the loss function, the loss may have numerous minimizers, however, when the model is under-parameterized, as in Figure 8, none of those minimizers would make the loss zero. Finding the same minimizer of training loss is equivalent to obtaining the same trained model, hence unlike the over-parameterized setting, the training regime is focused on finding the global minimizer of the training loss, i.e., finding the best shape for the decision boundary inside the convex hull.

As we increase the number of neurons with increments of 2, we see the model with 30 neurons can perfectly separate the 2 classes in our training set. Figure 9 shows the output of 3 different 30-neuron models that are trained with different training regimes.

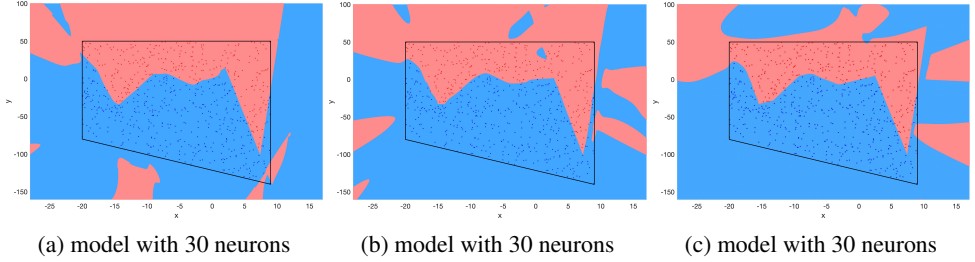

(a) model with 30 neurons      (b) model with 30 neurons      (c) model with 30 neurons

Figure 9: The case where the model has just about enough capacity to fit the training set. We see that based on the training regime, we can get slightly different decision boundaries inside the hull, all of which perfectly separate the 2 classes. The shape of decision boundaries outside the hull can also vary based on the training regime.

Finally, we consider models that are highly over-parameterized. In this regime, there are infinite number of parameter configurations that minimize the training loss to zero, which is equivalent to developing the decision boundaries that perfectly separate our two classes.

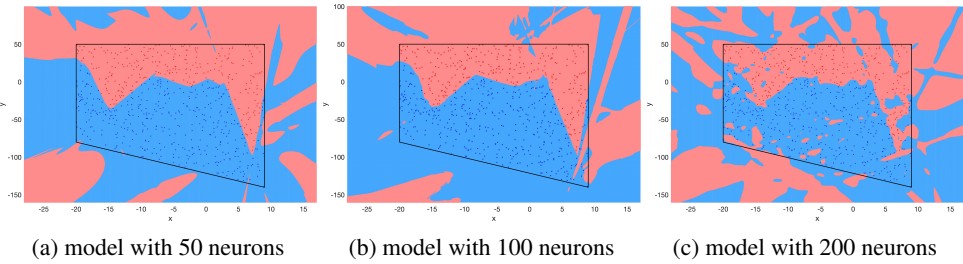

(a) model with 50 neurons      (b) model with 100 neurons      (c) model with 200 neurons

Figure 10: As we increase the degree of over-parameterization, from 50 neurons to 200, the number of disjoint decision boundaries increase, inside and outside the hull.

The above observations seem to explain why we need over-parameterized models for deep learning and also explain why the generalization of deep learning models are so susceptible to different training regimes. Appendix B provides further discussion.

## 5    Conclusion and future work

We showed that most of testing data for some standard image classification models lie outside the convex hull of training sets, both in pixel space and in wavelet space. Therefore, the generalization of a deep network partially relies on its capability to extrapolate outside the boundaries of the data it has seen during training. Based on this observation, the significant number of studies that focus on interpolation regimes seem to be insufficient to explain the generalization of deep networks.

From this perspective, over-parameterization of models may be considered a necessity to desirably form the extension of decision boundaries outside the convex hull of data. This can be proven for polynomial regression models using the orthogonal system of Legendre polynomials. Moreover, we showed that the training regime can significantly affect the shape of decision boundaries outside the convex hulls, affecting the accuracy of a model in its extrapolation. We investigated a 2-layer ReLU network and a polynomial decision boundary to demonstrate these ideas.

In future work, we plan to more closely analyze the effect of over-parameterization and training regimes on the shape of decision boundaries outside the convex hulls, and investigate how that affects the generalization. We also plan to study how sensitive the classifications of standard trained models are w.r.t the minimum perturbations that would bring testing images on the surface of $\mathcal{H}^{tr}$.

Projecting all testing samples to the surface of $\mathcal{H}^{tr}$ and investigating whether a model can learn to classify them would provide insights on how far we can rely on interpolation, and over-parameterization is still necessary in that setting.

Studying the convex hulls of internal representations of the data in a trained network is another direction that can be pursued. Such analysis can be performed, separately for each class in the dataset. It has been speculated that a given image classification dataset lies on a lower dimensional manifold and such manifold is what a deep learning model learns from the data. Study of convex hulls might provide insights about such manifold and also about the distribution of training and testing sets.

Finally, measuring the volume of the convex hulls of training and testing sets, their overlap, and also the volume of the domain that remains unoccupied may be insightful.

## Acknowledgments and Disclosure of Funding

R.Y. thanks Yaim Cooper for helpful discussions. R.Y. was supported by a fellowship from the Department of Veteran Affairs. The views expressed in this manuscript are those of the author and do not necessarily reflect the position or policy of the Department of Veterans Affairs or the United States government.

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

## A The case of random data in high-dimensional space

The question may arise that what happens if we have random data points in the same high-dimensional space. Would the data points still fall outside the $\mathcal{H}^{tr}$? Would the distance to $\mathcal{H}^{tr}$ be the same? The answers are yes and no, respectively.

To gain some insights, let's consider the CIFAR-10 datasets. We can randomly shuffle all the pixel values in all the images of training and testing sets. In such case, the shuffled testing data would still fall outside the $\mathcal{H}^{tr}$ of shuffled data. But, their distance to $\mathcal{H}^{tr}$ would be orders of magnitude larger.

Alternatively, if we generate random points in the same domain as the pixel space of CIFAR-10, (i.e., domain 3,072 dimensions bounded between 0 and 255), again, the testing data will be outside the $\mathcal{H}^{tr}$.[2] This time, the distance to $\mathcal{H}^{tr}$ would be much larger even compared to the case of shuffling the pixel values.

Therefore, we can conclude that with such number of training samples in such high-dimensional domains, one can expect the testing samples to be outside their $\mathcal{H}^{tr}$. However, the distance to the convex hulls are much closer for our image datasets, compared to random data points, because for each testing image, $x_i^{te}$, there are a group of training images that form a surface on the $\mathcal{H}^{tr}$, and that surface is much closer to $x_i^{te}$, compared to the surface that a set of random data points can create.

## B Extrapolation and interpolation in tandem

Earlier, we reported that the extent of extrapolation, for the CIFAR-10 dataset, is about 27% of the diameter of its $\mathcal{H}^{tr}$. We also reported that almost all the testing samples are outside the $\mathcal{H}^{tr}$. Based on these observations, we can say that the task of classifying the testing set of CIFAR-10 is extrapolation. But, how does the **interpolation** affect our ability to **extrapolate**? We can explain this using the decision boundaries of the models.

For image classification, our domain is a $d$-dimensional hyper-cube, because pixel values are bounded. $d$ is the number of pixels. The convex hull of training set, $\mathcal{H}^{tr}$, is a shape with $\leq d$ dimension, and sits somewhere in that hyper-cube. For CIFAR-10 dataset, $\mathcal{H}^{tr}$ has exactly $d$ dimensions. The testing samples sit around the $\mathcal{H}^{tr}$, like a mist, not too far, and not too close to $\mathcal{H}^{tr}$. The distance of testing samples from $\mathcal{H}^{tr}$ ranges between 1%-27% of its diameter, with average value of 10%.

As we mentioned earlier, a classification function/model partitions its domain and assigns a class to each of the partitions. The partitions are defined by decision boundaries, and so is the model. Basically, the training process partitions the shape $\mathcal{H}^{tr}$ by defining a finite set of decision boundaries inside it. We can define this process as interpolation, especially if we are only concerned about the location of decision boundaries in $\mathcal{H}^{tr}$. Some of the decision boundaries defined in this process will reach the surface of $\mathcal{H}^{tr}$ and extend outside it. Theses decision boundaries and their extensions outside the hull are the ones that a model relies upon in order to classify the images outside the hull.

Because the testing images are not too far outside the hull, the space to shape the extensions of decision boundaries is limited. Therefore, the locations where the decision boundaries reach the surface of $\mathcal{H}^{tr}$ is of great importance. Overall, interpolation and extrapolation work in tandem to shape the decision boundaries and the functional performance of image classification models.

Going back to the hyper-cube, during the training, the shape of $\mathcal{H}^{tr}$ is partitioned with some nonlinear surfaces (decision boundaries), and some of those surfaces extend outside the $\mathcal{H}^{tr}$. The locations where the decision boundaries reach the surface of $\mathcal{H}^{tr}$ and their extension outside the $\mathcal{H}^{tr}$ is critical in how the model classifies the testing images that are sitting around the $\mathcal{H}^{tr}$.

---

[2]This relates to the limit theorems for the convex hull of random points in higher dimensions [Hueter, 1999] and also to studies on separability and distribution of random points [Fink et al., 2016].

