# OpenReview forum: "Deep Learning Generalization and the Convex Hull of Training Sets"
_NeurIPS.cc/2020/Workshop/DL-IG — NeurIPSW 2020: DL-IG Poster_

### Official Review · AnonReviewer1 · 2020-10-22
**Review of "Deep Learning Generalization and the Convex Hull of Training Sets"**

**Rating:** 5
**Confidence:** 4

**Review:**

The set of natural images is often thought to lie on a low dimensional manifold embedded in the pixel space. The authors comment that "Specifically, we study the convex hulls of image classification datasets (both in the pixel space and in the wavelet space), and show that a considerable portion of testing images fall outside the convex hull of training sets." It seems to me that the convex hull in the original pixel space may not have any connection to the shape of this manifold -- some kind of convex hull on the manifold would be more correct. So the observation that test images lie outside the convex hull of train images does not seem very surprising to me. Why not look at the convex hull of the extracted features of the training set (such as the pre-final layer in a trained neural network)?

Unfortunately, the experiments in Section 3 and 4 did not convince me of the authors' point that "... over-parameterization of deep learning models is a necessity for shaping the extension of decision boundaries". In Section 4, the underparametrized model appears to find the "simplest" boundary. The overparametrized models need additional regularization to make the boundaries simple.

---

### Official Review · AnonReviewer2 · 2020-11-07

**Rating:** 5
**Confidence:** 4

**Review:**

This paper shows that a large fraction of the test data in typical datasets such as MNIST and CIFAR-10 lies outside the convex hull of the training set. The paper uses this to argue further that the decision boundary of the classifier that extends outside the convex hull of the training set should also accurately classify the test data and this implies that a model should be over-parametrized.

I am having trouble understanding this paper. I understand the experiment in Section 2 on computing the distance between the test images and the training images. Can you explain the result “if 26% of MNIST test images lie outside the convex hull but 87% of test images lie outside the convex hull after a wavelet transform”? MNIST is easier to classify using a linear classifier after the wavelet transform, so generalization is not just about test data being outside the hull but also where the test data are located. To impose some “prior“ (regularity would be a more appropriate word) on the decision boundary outside the convex hull, one needs an inductive bias. Mere over-parametrization can give many different decision boundaries, as shown in in Fig. 4 and therefore it is not clear whether the over-parametrized model should generalize well.

---

### Author Response · Authors · 2020-12-09
**Video for poster session**

Youtube video: https://youtu.be/xL9qG8PN4s8

---

### Decision · Program_Chairs · 2020-11-07

**Decision:**

Accept (Poster)

**Comment:**

Please incorporate the reviewer feedback into your manuscript.